# Joint CO<sub>2</sub> state and flux estimation with the 4D-Var system EURAD-IM

Johannes Klimpt <sup>1,2</sup>, Elmar Friese <sup>1</sup>, and Hendrik Elbern <sup>1,2</sup>

<sup>1</sup>Rhenish Institute for Environmental Research at the University of Cologne, 50931 Köln, Germany <sup>2</sup>Institute for Energy and Climate Research (IEK-8), Forschungszentrum Jülich, 52425 Jülich, Germany *Correspondence to:* J.Klimpt, j.klimpt@uni-koeln.de

Abstract. This idealized regional atmospheric inversion study assesses the potential of the 4-dimensional variational (4D-Var) method to estimate  $CO_2$  fluxes and the atmospheric  $CO_2$  concentration state jointly. In order to distinguish and quantify the surface-atmosphere  $CO_2$  fluxes, combining anthropogenic  $CO_2$  emissions, photosynthesis, and respiration, we include uncertainties of initial values, which arise from highly uncertain surface fluxes and night-time transport. Therefor a new calculation

- of the background error standard deviation for the  $CO_2$  fluxes was developed. To suppress spurious wiggles occurring from advection, an absolute monotone advection scheme with low numeric diffusion and its adjoint has been implemented. The inversion by the EURopean Air pollution Dispersion-Inverse Model (EURAD-IM) with 5 km resolution in Central Europe is validated by synthetic half hourly measurements from eleven concentration towers. A significant improvement of the analysis is shown if initial values and  $CO_2$  fluxes are optimised jointly, compared to optimising  $CO_2$  fluxes alone, without estimating
- uncertainty of atmospheric concentration. We find that joint estimation of carbon fluxes and initial states requires a careful balance of the background error covariance matrices but enables a more detailed analysis of atmospheric  $CO_2$  and the surface-atmosphere fluxes.

# 1 Introduction

The quantification of terrestrial CO<sub>2</sub> sinks and sources is important in order to understand the carbon cycle and its fundamental role in the global climate system. On a global scale, approximately 30% of the anthropogenic emissions are removed by oceans (Wanninkhof et al., 2013) as well as 30% by terrestrial biosphere (Sitch et al., 2013), the latter one showing much larger temporal and spatial variability (Jung et al., 2011).

One widely used method for the estimation of the surface-atmosphere  $CO_2$  exchange is atmospheric inverse modelling, using  $CO_2$  concentration measurements and atmospheric transport models. On a global scale atmospheric inversions improve

the quantification of natural and anthropogenic carbon fluxes (Tans et al., 1990; Enting et al., 1993; Rödenbeck et al., 2003; Rayner et al., 2008), although the uncertainty between studies is large, especially between continents (Gurney et al., 2002; Baker et al., 2006; Peters et al., 2010).

Locally the increase of spatio-temporal model resolution leads to even higher uncertainties of atmospheric inversion studies (Lauvaux et al., 2008; Broquet et al., 2011), mainly for two reasons: (1) A derivation of appropriate a priori surface carbon

fluxes is difficult due to spatial heterogeneity. In particular European land use is highly variable since large parts of the terrestrial biosphere are actively managed by humans and are often embedded in densely populated areas, including cities and industrial areas (Peters et al., 2010). (2) Atmospheric transport modelling introduces uncertainty to atmospheric  $CO_2$  mixing ratio of several ppm due to advection (Lin and Gerbig, 2005) and vertical mixing (Gerbig et al., 2008). Although uncertainty

5 from transport model errors is considerably lower than uncertainty from  $CO_2$  surface fluxes (Tolk et al., 2009), the skills of the transport model are essential for  $CO_2$  inversions.

Additionally to the mentioned difficulties of proper transport models and surface-atmosphere  $CO_2$  fluxes one major obstacle for  $CO_2$  inversions is the sparsity of representative concentration observations, e.g. Wu et al. (2016) addresses the question of optimal location of observations in inversion systems. To address uncertainty appropriately to the mentioned model processes and input data is among the most challenging issues for atmospheric  $CO_2$  inversions.

Tolk et al. (2011) compare the uncertainty reduction for different strategies of inverse carbon modelling, suggesting an optimisation of biosphere model parameters or a pixel inversion of respiration and photosynthesis on a regional scale.
To reduce uncertainty of a priori fluxes, additional carbon flux measurements from eddy covariance towers can be used .
Lauvaux et al. (2008), Cooley et al. (2013), and Zhu et al. (2014) combine this so-called bottom-up approach with top-down

- atmospheric inversion in order to quantify regional carbon fluxes. To reduce errors in atmospheric transport processes, different strategies are investigated. For example additional information from radon measurements (Broquet et al., 2011) or the energy balance (Tolk et al., 2009) are used to gain information of atmospheric CO<sub>2</sub> dispersion. Studies with an horizontal resolution of a few km (Ahmadov et al., 2007, 2009; Pillai et al., 2011) increase confidence in modelled CO<sub>2</sub> mixing ratios compared to coarser resolution, although comparisons of meso-
- scale models reveal discrepancies for the vertical boundary layer (Sarrat et al., 2007; Geels et al., 2007). Uncertainty of vertical model transport is considerably higher during night due to uncertain mixing height (Lauvaux et al., 2008; Dolman et al., 2009; Kretschmer et al., 2014), leading to a poor representativity of concentration during morning hours. Still, optimisation of initial conditions is often mitigated by long spin-up periods or is investigated at a coarse subspace (Peylin et al., 2005; Lauvaux et al., 2008).
- This study investigates how  $CO_2$  flux uncertainty from inversions can be reduced by a proper accounting of uncertainty of atmospheric concentration. The feasibility and limitations of a comprehensive 4D-Var inversion system to analyse anthropogenic emissions, photosynthesis, ecosystem respiration, and initial values at each grid cell jointly, are validated in this study. This paper is organised as follows. Section 2 delivers the theoretical background of the joint optimisation of fluxes and initial

initial values are presented and discussed in Sect. 4. Section 5 summarises the results.

values, Sect. 3 describes the implementation into the EURAD-IM. Numerical experiments validating the impact of erroneous

10

# 2 Theory for joint variational assimilation

In this study we want to optimise two parameters with the 4D-Var data assimilation method: initial values and flux factors jointly. While the proceeding of optimising initial values is well-known, we want to visualise the basic concept of optimising

**Figure 1.** The diurnal time profile of power generation given by Memmesheimer et al. (1995) and spatial averaged biogenic fluxes calculated with WRF-CLM at 24 July 2012 (absolute weights sum up to 1). An example with an analysed flux factor of 0.75 for photosynthesis is shown, decreasing the total amount but preserving the profile.

flux factors and joint optimisation. The main idea (Elbern et al., 2000) is to reduce the degree of freedom of the flux rate space by not optimising the fluxes at each time step  $t_i$ . Rather it is pointed out that, due to the better knowledge of the diurnal cycle of fluxes, one efficient parameter to optimise is their diurnal amplitude, as simulated by the forward model.

This study optimises three fluxes, anthropogenic emissions, photosynthesis, and respiration. For simplicity we will define the
flux space to be of the same dimension as the state space (ℝ<sup>n</sup>). The flux rate vector *f* ∈ ℝ<sup>n</sup> scales the a priori knowledge of the background flux U<sup>b</sup> by an optimisation factor per grid point and per flux type, while the relative diurnal profile variation remains unchanged. For notational convenience we define U<sup>b</sup> ∈ ℝ<sup>n×n</sup> to be a diagonal matrix.

The actual fluxes are thus (diag(a) designates a diagonal matrix with entries of the vector a)

$$\mathbf{U}_{i+1/2} = \operatorname{diag}(\mathbf{U}_{i+1/2}^{\mathsf{b}} \mathbf{f}) \in \mathbb{R}^{n \times n}, \quad \forall \ i = 0, \dots, N-1,$$

$$\tag{1}$$

10 with U<sub>i+1/2</sub> denoting the fluxes in the time interval [t<sub>i</sub>, t<sub>i+1</sub>] introduced instantaneously at t<sub>i+1/2</sub>. For notational convenience subindices in this paper will be used exclusively to identify discrete time. The simulated time period is defined as [t<sub>0</sub>, t<sub>1</sub>,...,t<sub>N</sub>], (·)<sub>i</sub> refers to t<sub>i</sub>, (·)<sub>i+1/2</sub> to (t<sub>i</sub> + t<sub>i+1</sub>)/2, and (·)<sub>i,j</sub> to the interval [t<sub>i</sub>, t<sub>j</sub>].

Figure 1 shows the diurnal cycle of the three  $CO_2$  fluxes, anthropogenic emissions, photosynthesis, and respiration, where the photosynthesis is exemplarily rescaled. The actual parameter of optimisation will be  $g := \ln(f)$  due to two reasons: (1) The

15 transformation results in analog partial costs for flux factors g and 1/g. (2) Since f > 0 it can be described by a log-normal distribution, resulting in a Gaussian probability density function for g. Fletcher and Zupanski (2006) and Fletcher (2010) describe 4D-Var systems with hybrid Gaussian and log-normal distribution in general.

The idea for the main advantage of joint optimisation of initial values and flux factors is depicted in Fig. 2. While the influence