# Peer review of "Joint CO2 state and flux estimation with the 4D-Var system EURAD-IM"

_Geoscientific Model Development, 2016_

## Short Comment (SC1) · 22 Aug 2016

Dear authors,

In my role as Executive editor of GMD, I would like to bring to your attention our Editorial version 1.1:

http://www.geosci-model-dev.net/8/3487/2015/gmd-8-3487-2015.html

This highlights some requirements of papers published in GMD, which is also available on the GMD website in the 'Manuscript Types' section:

http://www.geoscientific-model-development.net/submission/manuscript_types.html

In particular, please note that for your paper, the following requirement has not been met in the Discussions paper:

[Figure]

- "The main paper must give the model name and version number (or other unique identifier) in the title."

Please add a version number for EURAD-IM in the title upon your revised submission to GMD.

Yours,

Astrid Kerkweg

---

## Referee Comment (RC1) · Anonymous Referee #1 · 26 Oct 2016

The manuscript "Joint CO2 state and flux estimation with the 4D-Var system EURAD-IM" by Klimpt et al. describes a variational data assimilation system for the regional atmospheric inverse modeling of natural and anthropogenic surface CO2 fluxes. The study focuses on the asset of including the CO2 atmospheric transport model initial conditions in the optimization parameters. The demonstration of this asset is supported by academic twin experiments. The study spends time on properly deriving the equations underlying the data assimilation system, the transport model and its adjoint.

The text is often concise and clean. The inversion system presented in this study is clearly interesting and should have some potential for the CO2 regional atmospheric inversion activities.

It is difficult to assess how much the authors had to adapt to EURAD-IM system for the

specific application described in this study. However, even though I can assume that it required efforts and significant developments, I feel that the deficiencies of this study make it hardly acceptable for publication. I would encourage the authors to resubmit a new manuscript better focused on a more critical asset brought by their CO2 configuration of EURAD-IM compared to other CO2 (variational) regional inversion systems than the corrections to the initial condition. The paper should better emphasize the new developments they had to conduct in order to adapt the EURAD-IM system to CO2 flux inversion which would give the opportunity to highlight how much this activity differs from the monitoring of concentrations or emissions of other species. On the other hand, it should avoid deriving all the details of the common theory underlying the variational inverse modeling framework that can be found in other publications.

1) My main feeling about this study is that it misses some of the basic knowledge on the CO2 regional flux inversion even though it refers to relevant studies in the field:

- The most important point: the adjustment of the initial condition is not a critical issue for regional inverse modeling studies. The influence of the initial condition for a domain such as the area of interest in this study hardly persists after 1 day of simulation, while the windows of analysis for the atmospheric inversions dedicated to natural fluxes (see my point about anthropogenic sources below) generally range between 1 month and 1 year, and while such inversions aim at deriving precise estimates of the fluxes typically at the 1-month to annual scale. In such a context, running 12-h inversion experiments like in this study is not really relevant. While this paper clearly targets estimates of the fluxes, it is much influenced by sequential data assimilation systems which target the analysis and forecasting of the atmospheric concentrations (e.g. in air quality). The author may have failed to understand the critical difference between these two types of activities. A true critical source of errors when conducting regional atmospheric inversion are the model CO2 boundary conditions (even though these conditions are poorly discussed at line 23-24 p5 and even though they seem to have been ignored on figure 2). An increasing number of inverse modeling studies (e.g. Lauvaux et al. 2012,

Henne et al. 2016) include the model boundary conditions among the optimization parameters. Of note is, however, that the initial conditions play some role for global systems and that, consequently, the joint adjustment of the (global) model CO2 initial conditions and fluxes is not something new in the CO2 inverse modeling community (e.g. Chevallier et al. 2013).

- The choice of the measurement sites is not critical for twin experiments with synthetic data. However, the authors made an embarrassing mistake by selecting locations corresponding to eddy covariance flux measurement sites (from the eddy covariance flux database FLUXNET). Such sites measure high temporal variations of atmospheric CO2 to derive local flux estimates, but the accuracy of these CO2 measurements is not consistent with the requirement of atmospheric inversion. The confusion between the atmospheric CO2 measurement sites and the eddy covariance sites is also highlighted in table 1 by the column "Vegetation". This indication is relevant and usually given for the representativity of eddy covariance sites but not for the atmospheric sites dedicated to inversions, which should integrate the signal from large areas. The actual atmospheric CO2 measurement sites correspond to the CarboEurope, ICOS... networks (e.g. https://icos-atc.lsce.ipsl.fr/stations). Since some sites include both eddy covariance and atmospheric CO2 measurements, some of the atmospheric sites are included in Table 1. But some sites clearly correspond to locations where there is no well calibrated atmospheric data and, on the opposite, the list misses some of the iconic stations of the european network close or within the area of interest.

- The assumption that the system could solve for the anthropogenic emissions along with biogenic fluxes based on the type of observation network used in this study is problematic. it is wrong that "On a global scale atmospheric inversions improve the quantification of (natural and) anthropogenic carbon fluxes" and most of the state of the art global atmospheric inversions have been run assuming that the knowledge on the anthropogenic emissions from gridded inventories is perfect (Peylin et al. 2013). At the typical ICOS measurement stations that are located at a certain distance from

major anthropogenic CO2 point sources and which are dedicated to the natural fluxes, the signal from anthropogenic emissions is generally much lower than that from the biogenic fluxes (at least during the growing season) and far more difficult to deal with since anthropogenic emissions are far more localized and heterogeneous than biogenic emissions. The correlation of the uncertainty in these emissions is extremely difficult to characterize and the diffusion equations (l8-11 p11) do not seem adapted to model this correlation. Indeed, the authors have to distort (i.e. set-up to deal with concerns that are not related to assumptions on its actual structure: see points 1. to 3. at lines 7-9 on page 12 and more generally 3.8.2) the prior uncertainty covariance matrix to ensure that the system gets some potential to solve for the anthropogenic emissions.

- Adjusting the fluxes without solving for the diurnal cycle of the fluxes (at least solving for the nighttime and daytime fluxes separately) while assimilating data during daytime only could be seen as another issue of the system proposed in this study. There are high uncertainties in the diurnal cycle of all the types of fluxes that are solved for here (anthropogenic emissions, photosynthesis and respiration) opposed to what l2 on p3 seems to indicate. Regarding the limitation of the number of optimization parameters: other inverse modeling studies were conducted over time windows that are far larger than 12h, with a number of optimization parameters that is larger than the one considered here, even when including the 3D field of the CO2 initial conditions. So I am not sure to understand what are the computation limitations that prevent from solving for the diurnal cycle of the fluxes in this study.

- The ability of transport models to assimilate data at 6am is debatable. CO2 Regional inverse modeling systems generally use afternoon data only.

- The value of 2ppm (l11p13) given to the observation error, which has to account for transport modeling error, seems too small compared to the typical CO2 variability simulated and measured at CO2 atmospheric sites in Europe.

- I am surprised to read that in the region of interest, most of the anthropogenic emissions are connected to "few power plants" (l17 p13). Could the authors display the map of the anthropogenic emissions they use in Figure 4 ? or maybe show the zoom on the region of interest (for all types of fluxes).

2) Some general comments by sections:

The abstract and introduction (and to some extent sections 2 and 3 too) fails to give a precise framework and concrete objective for the study or for the system it uses. In the abstract, sentences like l4 to 6 p1 are given out of context and impossible to understand. A particular issue is that the introduction (and sections 2 and 3) fails to give an idea on the temporal and spatial scales corresponding to the study (the scales of interest when analyzing the fluxes, the typical inversion time window...), which prevents from weighting the discussions regarding the initial conditions in this sections (see my general comment above on that topic). On the other hand, the introduction goes in various directions and raises a lot of different topics (on the transport modeling errors, the spatial heterogeneity of fluxes in Europe, the development of coupled ecosystem - transport models that can assimilate both eddy covariance flux measurements and $CO_2$ atmospheric data...) that are associated with atmospheric inversion but that are not be strongly connected to this study. The central topic of the initial conditions is introduced quite abruptly at line 22p2.

Section 2 and most of Section 3 are dedicated to the mathematics that are generally quite common for atmospheric inversion (or more generally 4DVAR incremental data assimilation), transport modeling and adjoint transport modeling: in general, the equations correspond to many of the systems that have already been used in atmospheric inversion. I may be wrong but I also feel that the main pieces of the CO2 EURAD-IM system in this study (transport model, adjoint, cost function optimization scheme, joint optimization of initial conditions and surface fluxes etc.) were already in place for previous studies with EURAD-IM dedicated to other species. Therefore I am not convinced that it is useful to have all this theoretical framework in this paper.

Section 4 does not discuss the consistency between the set-up of B and K and the perturbation applied to the initial state and flux factors in the twin experiments. Comments on figure 6 at l22-23 p14 are a bit optimistic : the overestimation of the initial concentration in the footprint of the measurements sites (up to -3 ppm) is often larger that the background underestimation (2 ppm). I do not understand l27-30 p14 and the story about the wiggles.

3) A lot of parts of the text are quite confusing or misleading e.g. (I cannot list all issues) - in the abstract: "uncertainties of initial values, which arise from highly uncertain surface fluxes and night-time transport" - the 3rd paragraph of the introduction dedicated to transport error seems to say that transport errors are larger with regional / mesoscale transport models than with global transport models - "accounting of uncertainty of atmospheric concentration" at line 24-25 p2 - "validating the impact of erroneous initial values" line 29-30 p2 - l5p3 and in general : the story about the dimension of the flux space vs. initial concentration space is really misleading, given that, even if accounting for injection heights for the anthropogenic emissions (i.e. using a 3D field in space to describe these emissions), and even if considering the 12h inversion time window of section 4 for which the system does not solve for the flux temporal variations, there are three types of flux fields vs. one initial concentration field so that the two dimensions are strongly different. I feel that using the same letter for the two dimensions does not significantly simplify things, so I do not understand why the authors do it. - the two last lines of p6: what do they mean ? which assumptions are used to select the model vertical layer in which to locate the stations ? . . .

References:

Chevallier, F.: On the parallelization of atmospheric inversions of CO2 surface fluxes within a variational framework, Geosci. Model Dev., 6, 783-790, doi:10.5194/gmd-6-783-2013, 2013.

Henne, S., Brunner, D., Oney, B., Leuenberger, M., Eugster, W., Bamberger, I., Meinhardt, F., Steinbacher, M., and Emmenegger, L.: Validation of the Swiss methane emission inventory by atmospheric observations and inverse modelling, Atmos. Chem. Phys., 16, 3683-3710, doi:10.5194/acp-16-3683-2016, 2016.

Lauvaux, T., Schuh, A. E., Uliasz, M., Richardson, S., Miles, N., Andrews, A. E., Sweeney, C., Diaz, L. I., Martins, D., Shepson, P. B., and Davis, K. J.: Constraining the CO2 budget of the corn belt: exploring uncertainties from the assumptions in a mesoscale inverse system, Atmos. Chem. Phys., 12, 337-354, doi:10.5194/acp-12-337-2012, 2012.

Peylin, P., Law, R. M., Gurney, K. R., Chevallier, F., Jacobson, A. R., Maki, T., Niwa, Y., Patra, P. K., Peters, W., Rayner, P. J., Rödenbeck, C., van der Laan-Luijkx, I. T., and Zhang, X.: Global atmospheric carbon budget: results from an ensemble of atmospheric CO2 inversions, Biogeosciences, 10, 6699-6720, doi:10.5194/bg-10-6699-2013, 2013.

---

## Referee Comment (RC2) · P. Rayner (Referee) · 8 Nov 2016

This paper considers the joint assimilation of emissions of CO2 and the initial condition of CO2 concentration. It tests assimilations with and without the initial condition in a standard Observing System simulation Experiment case. It shows that, over the short assimilation window it considers, the inclusion of the initial condition improves the assimilation of fluxes, the usual target for such problems. It requires careful balancing of the prior uncertainty covariances for the unknowns. The paper describes a potentially important technical advance in an area of geophysical modelling and data analysis and so is certainly within scope for GMD.

That said I do not believe the manuscript is suitable for publication at the moment. I see two serious problems with it.

[Figure]

First, the experiment is not a good reflection of how we perform inversions for surface fluxes in practice. In particular it is unusual to use such short assimilation windows even for regional inversions. This is important since the results here show that the effect of the initial condition is limited to the start even of this assimilation window. If we used the more normal practice in the inversion community of running for a week, a month or a year, it is hard to see that the effect would be important. The cited paper from Peylin et al. (2005) already suggested this with an initial condition affecting the first few days. That study was for a much larger domain where we could expect the flushing time for the initial concentration to be much larger.

It is perhaps unfair to criticize authors for something their paper does not do but there is a related problem in regional inverse studies which is, I think, very important indeed. The impact of lateral boundaries, here highly simplified, becomes more and more important as the domain shrinks. I believe exposing this to an assimilation system will have a much larger impact than the initial condition.

There is also a related problem where I think the joint assimilation is important. This is the analysis of the atmospheric chemical state. Here the variable of interest is the concentration while the emissions are a nuisance variable. Such analyses aren't very interesting for $CO_2$ but are of great importance for other species.

My second concern is the treatment of prior covariances. It is twofold. I think the authors have borrowed too much from the methodology of numerical weather prediction. In NWP, the ultimate test of a good system is a good forecast and it is permissible to do what is necessary to get it. In a more elevated sense we could assume that the statistics of the system are well-tuned if the analyzed state is good. The $CO_2$ inverse case is not quite like that. We generally do not have some exterior metric we can test against. We are forced more to rely on analysis of our prior information in its own right. Thus questions of what the statistics of prior and observational covariances mean and how to verify them become crucial. Various papers from Chevallier and his group, for example, compare the prior flux in an inversion with independent pointwise data and

generate statistics accordingly.

Secondly, although I believe the covariance model employed is suitable for the initial condition, I do not understand its relevance for the emission covariance. Maybe I misread this but I do not see it as evolving according to some advective-diffusive process.

For these reasons I do not believe a paper along the lines of this manuscript will make a substantial contribution. I do believe that the approaches have value when applied to a range of related problems.

I have a series of more specific comments on the paper but I do not believe it is efficient for the authors to correct these until they can satisfy the editor that the overall paper should proceed.